# BERT Loses Patience:
# Fast and Robust Inference with Early Exit

**Wangchunshu Zhou**[1,*] **Canwen Xu**[2*]**, Tao Ge**[3]**, Julian McAuley**[2]**, Ke Xu**[1]**, Furu Wei**[3]
[1]Beihang University [2]University of California, San Diego [3]Microsoft Research Asia
[1]zhouwangchunshu@buaa.edu.cn,kexu@nlsde.buaa.edu.cn
[2]{cxu,jmcauley}@ucsd.edu [3]{tage,fuwei}@microsoft.com

## Abstract

In this paper, we propose Patience-based Early Exit, a straightforward yet effective inference method that can be used as a plug-and-play technique to simultaneously improve the efficiency and robustness of a pretrained language model (PLM). To achieve this, our approach couples an internal-classifier with each layer of a PLM and dynamically stops inference when the intermediate predictions of the internal classifiers remain unchanged for a pre-defined number of steps. Our approach improves inference efficiency as it allows the model to make a prediction with fewer layers. Meanwhile, experimental results with an ALBERT model show that our method can improve the accuracy and robustness of the model by preventing it from overthinking and exploiting multiple classifiers for prediction, yielding a better accuracy-speed trade-off compared to existing early exit methods.[2]

## 1 Introduction

In Natural Language Processing (NLP), pretraining and fine-tuning have become a new norm for many tasks. Pretrained language models (PLMs) (e.g., BERT [1], XLNet [2], RoBERTa [3], ALBERT [4]) contain many layers and millions or even billions of parameters, making them computationally expensive and inefficient regarding both memory consumption and latency. This drawback hinders their application in scenarios where inference speed and computational costs are crucial. Another bottleneck of overparameterized PLMs that stack dozens of Transformer layers is the "overthinking" problem [5] during their decision-making process. That is, for many input samples, their shallow representations at an earlier layer are adequate to make a correct classification, whereas the representations in the final layer may be otherwise distracted by over-complicated or irrelevant features that do not generalize well. The overthinking problem in PLMs leads to wasted computation, hinders model generalization, and may also make them vulnerable to adversarial attacks [6].

In this paper, we propose a novel **Pa**tience-**b**ased **E**arly **E**xit (PABEE) mechanism to enable models to stop inference dynamically. PABEE is inspired by the widely used Early Stopping [7, 8] strategy for model training. It enables better input-adaptive inference of PLMs to address the aforementioned limitations. Specifically, our approach couples an internal classifier with each layer of a PLM and dynamically stops inference when the intermediate predictions of the internal classifiers remain unchanged for $t$ times consecutively (see Figure 1b), where $t$ is a pre-defined patience. We first show that our method is able to improve the accuracy compared to conventional inference under certain assumptions. Then we conduct extensive experiments on the GLUE benchmark and show that PABEE outperforms existing prediction probability distribution-based exit criteria by a large margin. In addition, PABEE can simultaneously improve inference speed and adversarial robustness of the original

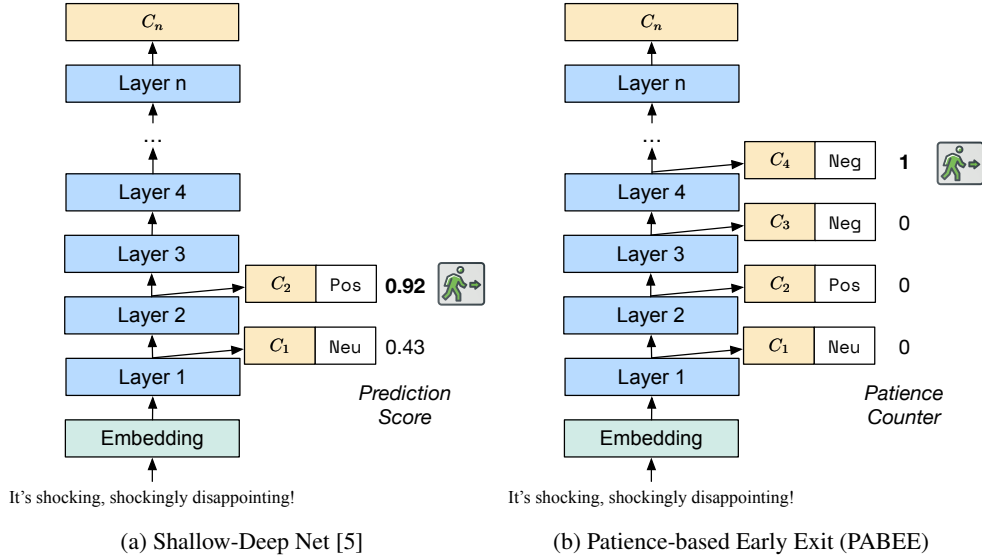

(a) Shallow-Deep Net [5]   (b) Patience-based Early Exit (PABEE)

Figure 1: Comparison between Shallow-Deep Net, a prediction score based early exit (threshold is set to 0.9), and our Patience-based Early Exit (patience $t = 1$). A classifier is denoted by $C_i$, and $n$ is the number of layers in a model. In this figure, Shallow-Deep incorrectly exits based on the prediction score while PABEE considers multiple classifiers and exits with a correct prediction.

model while retaining or even improving its original accuracy with minor additional effort in terms of model size and training time. Also, our method can dynamically adjust the accuracy-efficiency trade-off to fit different devices and resource constraints by tuning the patience hyperparameter without retraining the model, which is favored in real-world applications [9]. Although we focus on PLM in this paper, we also have conducted experiments on image classification tasks with the popular ResNet [10] as the backbone model and present the results in Appendix A to verify the generalization ability of PABEE.

To summarize, our contribution is two-fold: (1) We propose Patience-based Early Exit, a novel and effective inference mechanism and show its feasibility of improving the efficiency and the accuracy of deep neural networks with theoretical analysis. (2) Our empirical results on the GLUE benchmark highlight that our approach can simultaneously improve the accuracy and robustness of a competitive ALBERT model, while speeding up inference across different tasks with trivial additional training resources in terms of both time and parameters.

## 2   Related Work

Existing research in improving the efficiency of deep neural networks can be categorized into two streams: (1) *Static* approaches design compact models or compress heavy models, while the models remain static for all instances at inference (i.e., the input goes through the same layers); (2) *Dynamic* approaches allow the model to choose different computational paths according to different instances when doing inference. In this way, the simpler inputs usually require less calculation to make predictions. Our proposed PABEE falls into the second category.

**Static Approaches: Compact Network Design and Model Compression**   Many lightweight neural network architectures have been specifically designed for resource-constrained applications, including MobileNet [11], ShuffleNet [12], EfficientNet [13], and ALBERT [4], to name a few. For model compression, Han et al. [14] first proposed to sparsify deep models by removing non-significant synapses and then re-training to restore performance. Weight Quantization [15] and Knowledge Distillation [16] have also proved to be effective for compressing neural models. Recently, existing studies employ Knowledge Distillation [17–19], Weight Pruning [20–22] and Module Replacing [23] to accelerate PLMs.

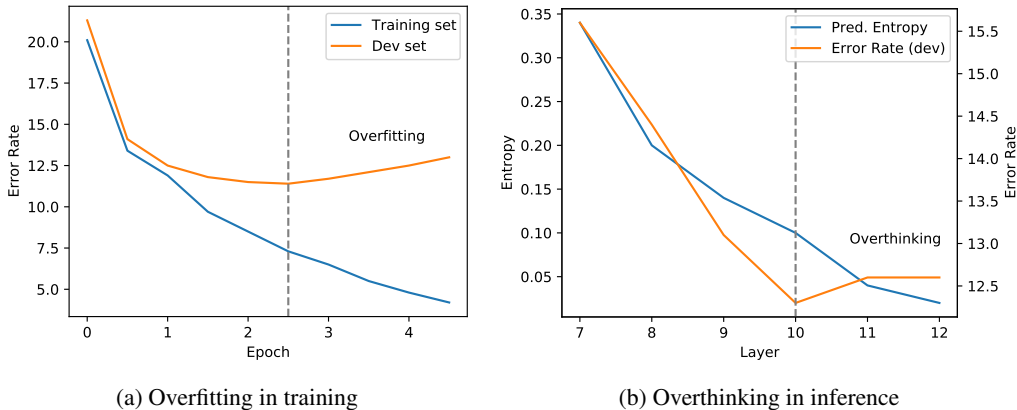

(a) Overfitting in training          (b) Overthinking in inference

Figure 2: Analogy between overfitting in training and overthinking in inference. **(a)** In training, the error rate keeps going down on the training set but goes up later on the development set. **(b)** We insert a classifier after every layer. Similarly, the predicted entropy keeps dropping when more layers are added to inference but the error rate goes up after 10 layers. The results are obtained with ALBERT-base on MRPC.

**Dynamic Approaches: Input-Adaptive Inference**     A parallel line of research for improving the efficiency of neural networks is to enable adaptive inference for various input instances. Adaptive Computation Time [24, 25] proposed to use a trainable halting mechanism to perform input-adaptive inference. However, training the halting model requires extra effort and also introduces additional parameters and inference cost. To alleviate this problem, BranchyNet [26] calculated the entropy of the prediction probability distribution as a proxy for the confidence of branch classifiers to enable early exit. Shallow-Deep Nets [5] leveraged the softmax scores of predictions of branch classifiers to mitigate the overthinking problem of DNNs. More recently, Hu et al. [27] leveraged this approach in adversarial training to improve the adversarial robustness of DNNs. In addition, existing approaches [24, 28] trained separate models to determine passing through or skipping each layer. Very recently, FastBERT [29] and DeeBERT [30] adapted confidence-based BranchyNet [26] for PLMs while RightTool [31] leveraged the same early-exit criterion as in the Shallow-Deep Network [5].

However, Schwartz et al. [31] recently revealed that prediction probability based methods often lead to substantial performance drop compared to an oracle that identifies the smallest model needed to solve a given instance. In addition, these methods only support classification and leave out regression, which limits their applications. Different from the recent work that directly employs existing efficient inference methods on top of PLMs, PABEE is a novel early-exit criterion that captures the inner-agreement between earlier and later internal classifiers and exploit multiple classifiers for inference, leading to better accuracy.

## 3 Patience-based Early Exit

Patience-based Early Exit (PABEE) is a plug-and-play method that can work well with minimal adjustment on training.

### 3.1 Motivation

We first conduct experiments to investigate the overthinking problem in PLMs. As shown in Figure 2b, we illustrate the prediction distribution entropy [26] and the error rate of the model on the development set as more layers join the prediction. Although the model becomes more "confident" (lower entropy indicates higher confidence in BranchyNet [26]) with its prediction as more layers join, the actual error rate instead increases after 10 layers. This phenomenon was discovered and named "overthinking" by Kaya et al. [5]. Similarly, as shown in Figure 2a, after 2.5 epochs of training, the model continues to get better accuracy on the training set but begins to deteriorate on the development set. This is the well-known overfitting problem which can be resolved by applying an early stopping

mechanism [7, 8]. From this aspect, overfitting in training and overthinking in inference are naturally alike, inspiring us to adopt an approach similar to early stopping for inference.

## 3.2 Inference

The inference process of PABEE is illustrated in Figure 1b. Formally, we define a common inference process as the input instance $\mathbf{x}$ goes through layers $L_1 \ldots L_n$ and the classifier/regressor $C_n$ to predict a class label distribution $\mathbf{y}$ (for classification) or a value $y$ (for regression, we assume the output dimension is 1 for brevity). We couple an internal classifier/regressor $C_1 \ldots C_{n-1}$ with each layer of $L_1 \ldots L_{n-1}$, respectively. For each layer $L_i$, we first calculate its hidden state $\mathbf{h}_i$:

$$\begin{aligned} \mathbf{h}_i &= L_i(\mathbf{h}_{i-1}) \\ \mathbf{h}_0 &= \text{Embedding}(\mathbf{x}) \end{aligned} \tag{1}$$

Then, we use its internal classifier/regressor to output a distribution or value as a per-layer prediction $\mathbf{y}_i = C_i(\mathbf{h}_i)$ or $y_i = C_i(\mathbf{h}_i)$. We use a counter $cnt$ to store the number of times that the predictions remain "unchanged". For classification, $cnt_i$ is calculated by:

$$cnt_i = \begin{cases} cnt_{i-1} + 1 & \arg\max(\mathbf{y}_i) = \arg\max(\mathbf{y}_{i-1}), \\ 0 & \arg\max(\mathbf{y}_i) \neq \arg\max(\mathbf{y}_{i-1}) \vee i = 0. \end{cases} \tag{2}$$

While for regression, $cnt_i$ is calculated by:

$$cnt_i = \begin{cases} cnt_{i-1} + 1 & |y_i - y_{i-1}| < \tau, \\ 0 & |y_i - y_{i-1}| \geq \tau \vee i = 0. \end{cases} \tag{3}$$

where $\tau$ is a pre-defined threshold. We stop inference early at layer $L_j$ when $cnt_j = t$. If this condition is never fulfilled, we use the final classifier $C_n$ for prediction. In this way, the model can exit early without passing through all layers to make a prediction.

As shown in Figure 1a, prediction score-based early exit relies on the softmax score. As revealed by prior work [32, 33], prediction of probability distributions (i.e., softmax scores) suffers from being over-confident to one class, making it an unreliable metric to represent confidence. Nevertheless, the capability of a low layer may not match its high confidence score. In Figure 1a, the second classifier outputs a high confidence score and incorrectly terminates inference. With Patience-based Early Exit, the stopping criteria is in a cross-layer fashion, preventing errors from one single classifier. Also, since PABEE comprehensively considers results from multiple classifiers, it can also benefit from an ensemble learning [34] effect.

## 3.3 Training

PABEE requires that we train internal classifiers to predict based on their corresponding layers' hidden states. For classification, the loss function $\mathcal{L}_i$ for classifier $C_i$ is calculated with cross entropy:

$$\mathcal{L}_i = -\sum_{z \in Z} \left[ \mathbb{1}\left[\mathbf{y}_i = z\right] \cdot \log P\left(\mathbf{y}_i = z | \mathbf{h}_i\right) \right] \tag{4}$$

where $z$ and $Z$ denote a class label and the set of class labels, respectively. For regression, the loss is instead calculated by a (mean) squared error:

$$\mathcal{L}_i = (y_i - \hat{y}_i)^2 \tag{5}$$

where $\hat{y}$ is the ground truth. Then, we calculate and train the model to minimize the total loss $\mathcal{L}$ by a weighted average following Kaya et al. [5]:

$$\mathcal{L} = \frac{\sum_{j=1}^{n} j \cdot \mathcal{L}_j}{\sum_{j=1}^{n} j} \tag{6}$$

In this way, every possible inference branch has been covered in the training process. Also, the weighted average can correspond to the relative inference cost of each internal classifier.

### 3.4 Theoretical Analysis

It is straightforward to see that Patience-based Early Exit is able to reduce inference latency. To understand whether and under what conditions it can also improve accuracy, we conduct a theoretical comparison of a model's accuracy with and without PABEE under a simplified condition. We consider the case of binary classification for simplicity and conclude that:

**Theorem 1** *Assuming the patience of PABEE inference is $t$, the total number of internal classifiers (IC) is $n$, the misclassification probability (i.e., error rate) of all internal classifiers (excluding the final classifier) is $q$, and the misclassification probability of the final classifier and the original classifier (without ICs) is $p$. Then the PABEE mechanism improves the accuracy of conventional inference as long as $n - t < (\frac{1}{2q})^t(\frac{p}{q}) - p$*

(the proof is detailed in Appendix B).

We can see the above inequality can be easily satisfied. For instance, when $n = 12$, $q = 0.2$, and $p = 0.1$, the above equation is satisfied as long as the patience $t \geq 4$. However, it is notable that assuming the accuracy of each internal classifiers to be equal and independent is generally not attainable in practice. Additionally, we verify the statistical feasibility of PABEE with Monte Carlo simulation in Appendix C. To further test PABEE with real data and tasks, we also conduct extensive experiments in the following section.

## 4 Experiments

### 4.1 Tasks and Datasets

We evaluate our proposed approach on the GLUE benchmark [35]. Specifically, we test on Microsoft Research Paraphrase Matching (MRPC) [36], Quora Question Pairs (QQP)[3] and STS-B [37] for Paraphrase Similarity Matching; Stanford Sentiment Treebank (SST-2) [38] for Sentiment Classification; Multi-Genre Natural Language Inference Matched (MNLI-m), Multi-Genre Natural Language Inference Mismatched (MNLI-mm) [39], Question Natural Language Inference (QNLI) [40] and Recognizing Textual Entailment (RTE) [35] for the Natural Language Inference (NLI) task; The Corpus of Linguistic Acceptability (CoLA) [41] for Linguistic Acceptability. We exclude WNLI [42] from GLUE following previous work [1, 19, 23]. For datasets with more than one metric, we report the arithmetic mean of the metrics.

### 4.2 Baselines

For GLUE tasks, we compare our approach with four types of baselines: **(1) Backbone models:** We choose ALBERT-base and BERT-base, which have approximately the same inference latency and accuracy. **(2) Directly reducing layers:** We experiment with the first 6 and 9 layers of the original (AL)BERT with a single output layer on the top, denoted by (AL)BERT-6L and (AL)BERT-9L, respectively. These two baselines help to set a lower bound for methods that do not employ any technique. **(3) Static model compression approaches:** For pruning, we include the results of Layer-Drop [22] and attention head pruning [20] on ALBERT. For reference, we also report the performance of state-of-the-art methods on compressing the BERT-base model with knowledge distillation or module replacing, including DistillBERT [17], BERT-PKD [18] and BERT-of-Theseus [23]. **(4) Input-adaptive inference:** Following the settings in concurrent studies [31, 29, 30], we add internal classifiers after each layer and apply different early exit criteria, including that employed by BranchyNet [26] and Shallow-Deep [5]. To make a fair comparison, the internal classifiers and their insertions are exactly same in both baselines and Patience-based Early Exit. We search over a set of thresholds to find the one delivering the best accuracy for the baselines while targeting a speed-up ratio between $1.30\times$ and $1.96\times$ (the speed-up ratios of (AL)BERT-9L and -6L, respectively).

### 4.3 Experimental Setting

**Training** We add a linear output layer after each intermediate layer of the pretrained BERT/ALBERT model as the internal classifiers. We perform grid search over batch sizes of

Table 1: Experimental results (median of 5 runs) of models with ALBERT backbone on the development set and the test set of GLUE. The numbers under each dataset indicate the number of training samples. The acceleration ratio is averaged across 8 tasks. We mark "-" on STS-B for BranchyNet and Shallow-Deep since they do not support regression.

| Method | #Param | Speed -up | CoLA (8.5K) | MNLI (393K) | MRPC (3.7K) | QNLI (105K) | QQP (364K) | RTE (2.5K) | SST-2 (67K) | STS-B (5.7K) | Macro Score |
|---|---|---|---|---|---|---|---|---|---|---|---|
| | | | | | | *Dev. Set* | | | | | |
| ALBERT-base [4] | 12M | 1.00× | 58.9 | 84.6 | 89.5 | 91.7 | 89.6 | 78.6 | 92.8 | 89.5 | 84.4 |
| ALBERT-6L | 12M | 1.96× | 53.4 | 80.2 | 85.8 | 87.2 | 86.8 | 73.6 | 89.8 | 83.4 | 80.0 |
| ALBERT-9L | 12M | 1.30× | 55.2 | 81.2 | 87.1 | 88.7 | 88.3 | 75.9 | 91.3 | 87.1 | 81.9 |
| LayerDrop [22] | 12M | 1.96× | 53.6 | 79.8 | 85.9 | 87.0 | 87.3 | 74.3 | 90.7 | 86.5 | 80.6 |
| HeadPrune [20] | 12M | 1.22× | 54.1 | 80.3 | 86.2 | 86.8 | 88.0 | 75.1 | 90.5 | 87.4 | 81.1 |
| BranchyNet [26] | 12M | 1.88× | 55.2 | 81.7 | 87.2 | 88.9 | 87.4 | 75.4 | 91.6 | - | - |
| Shallow-Deep [5] | 12M | 1.95× | 55.5 | 81.5 | 87.1 | 89.2 | 87.8 | 75.2 | 91.7 | - | - |
| PABEE *(ours)* | 12M | 1.57× | **61.2** | **85.1** | **90.0** | **91.8** | **89.6** | **80.1** | **93.0** | **90.1** | **85.1** |
| | | | | | | *Test Set* | | | | | |
| ALBERT-base [4] | 12M | 1.00× | 54.1 | 84.3 | 87.0 | 90.8 | 71.1 | 76.4 | 94.1 | 85.5 | 80.4 |
| PABEE *(ours)* | 12M | 1.57× | **55.7** | **84.8** | **87.4** | **91.0** | **71.2** | **77.3** | 94.1 | **85.7** | **80.9** |

{16, 32, 128}, and learning rates of {1e-5, 2e-5, 3e-5, 5e-5} with an Adam optimizer. We apply an early stopping mechanism and select the model with the best performance on the development set. We implement PABEE on the base of Hugging Face's Transformers [43]. We conduct our experiments on a single Nvidia V100 16GB GPU.

**Inference** Following prior work on input-adaptive inference [26, 5], inference is on a per-instance basis, i.e., the batch size for inference is set to 1. This is a common latency-sensitive production scenario when processing individual requests from different users [31]. We report the median performance over 5 runs with different random seeds because the performance on relatively small datasets such as CoLA and RTE usually has large variance. For PABEE, we set the patience $t = 6$ in the overall comparison to keep the speed-up ratio between $1.30\times$ and $1.96\times$ while obtaining good performance following Figure 4. We further analyze the behavior of the PABEE mechanism with different patience settings in Section 4.5.

Table 2: Experimental results (median of 5 runs) of BERT based models on the development set of GLUE. We mark "-" on STS-B for BranchyNet and Shallow-Deep since they do not support regression.

| Method | #Param | Speed -up | MNLI (393K) | SST-2 (67K) | STS-B (5.7K) |
|---|---|---|---|---|---|
| BERT-base [1] | 108M | 1.00× | 84.5 | 92.1 | 88.9 |
| BERT-6L | 66M | 1.96× | 80.1 | 89.6 | 81.2 |
| BERT-9L | 87M | 1.30× | 81.4 | 90.5 | 85.0 |
| DistilBERT [17] | 66M | 1.96× | 79.0 | 90.7 | 81.2 |
| BERT-PKD [23] | 66M | 1.96× | 81.3 | 91.3 | 86.2 |
| BERT-of-Theseus [23] | 66M | 1.96× | 82.3 | 91.5 | **88.7** |
| BranchyNet [26] | 108M | 1.87× | 80.3 | 90.4 | - |
| Shallow-Deep [5] | 108M | 1.91× | 80.5 | 90.6 | - |
| PABEE *(ours)* | 108M | 1.62× | **83.6** | **92.0** | **88.7** |

Table 3: Parameter numbers and training time (in minutes) until the best performing checkpoint (on the development set) with and without PABEE on ALBERT and BERT as backbone models.

| Method | #Param | | Train. time (min) | |
|---|---|---|---|---|
| | MNLI | SST-2 | MNLI | SST-2 |
| **ALBERT** | | | | |
| w/o PABEE | 12M | 12M | 234 | 113 |
| w/ PABEE | +36K | +24K | **227** | **108** |
| **BERT** | | | | |
| w/o PABEE | 108M | 108M | 247 | 121 |
| w/ PABEE | +36K | +24K | **242** | **120** |

## 4.4 Overall Comparison

We first report our main result on GLUE with ALBERT as the backbone model in Table 1. This choice is made because: (1) ALBERT is a state-of-the-art PLM for natural language understanding. (2) ALBERT is already very efficient in terms of the number of parameters and memory use because of its layer sharing mechanism, but still suffers from the problem of high inference latency. We can see that our approach outperforms all compared approaches on improving the inference efficiency of PLMs, demonstrating the effectiveness of the proposed PABEE mechanism. Surprisingly, our

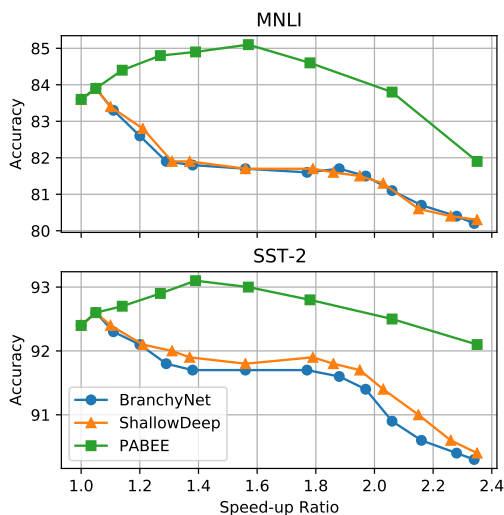

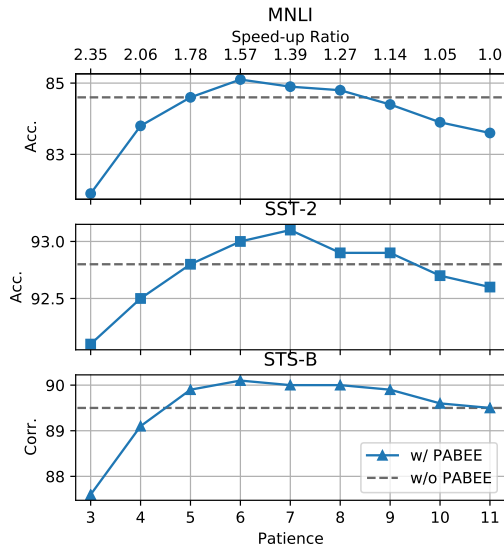

Figure 3: Speed-accuracy curves of BranchyNet, Shallow-Deep and PABEE on MNLI and SST-2 with ALBERT-base model.

Figure 4: Accuracy scores and speed-up ratios under different patience with ALBERT-base model. The baseline is denoted with gray dash lines.

approach consistently improves the performance of the original ALBERT model by a relatively large margin while speeding-up inference by $1.57\times$. This is, to the best of our knowledge, the first inference strategy that can improve both the speed and performance of a fine-tuned PLM.

To better compare the efficiency of PABEE with the method employed in BranchyNet and Shallow-Deep, we illustrate speed-accuracy curves in Figure 3 with different trade-off hyperparameters (i.e., threshold for BranchyNet and Shallow-Deep, patience for PABEE). Notably, PABEE retains higher accuracy than BranchyNet and Shallow-Deep under the same speed-up ratio, showing its superiority over prediction score based methods.

To demonstrate the versatility of our method with different PLMs, we report the results on a representative subset of GLUE with BERT [1] as the backbone model in Table 2. We can see that our BERT-based model significantly outperforms other compared methods with either knowledge distillation or prediction probability based input-adaptive inference methods. Notably, the performance is slightly lower than the original BERT model while PABEE improves the accuracy on ALBERT. We suspect that this is because the intermediate layers of BERT have never been connected to an output layer during pretraining, which leads to a mismatch between pretraining and fine-tuning when adding the internal classifiers. However, PABEE still has a higher accuracy than various knowledge distillation-based approaches as well as prediction probability distribution based models, showing its potential as a generic method for deep neural networks of different kinds.

As for the cost of training, we present parameter numbers and training time with and without PABEE with both BERT and ALBERT backbones in Table 3. Although more classifiers need to be trained, training PABEE is no slower (even slightly faster) than conventional fine-tuning, which may be attributed to the additional loss functions of added internal classifiers. This makes our approach appealing compared with other approaches for accelerating inference such as pruning or distillation because they require separately training another model for each speed-up ratio in addition to training the full model. Also, PABEE only introduces fewer than 40K parameters ($0.33\%$ of the original 12M parameters).

## 4.5 Analysis

**Impact of Patience** As illustrated in Figure 4, different patience can lead to different speed-up ratios and performance. For a 12-layer ALBERT model, PABEE reaches peak performance with a patience of 6 or 7. On MNLI, SST-2 and STS-B, PABEE can always outperform the baseline with patience between 5 and 8. Notably, unlike BranchyNet and Shallow-Deep, whose accuracy

Table 4: Experimental results (median of 5 runs) of different sizes of ALBERT on GLUE development set.

| Method | #Param | #Layer | Speed -up | MNLI (393K) | SST-2 (67K) | STS-B (5.7K) |
|---|---|---|---|---|---|---|
| ALBERT-base [4] | 12M | 12 | 1.00× | 84.6 | 92.8 | 89.5 |
| + PABEE | 12M | 12 | 1.57× | **85.1** | **93.0** | **90.1** |
| ALBERT-large [4] | 18M | 24 | 1.00× | 86.4 | 94.9 | 90.4 |
| + PABEE | 18M | 24 | 2.42× | **86.8** | **95.2** | **90.6** |

Table 5: Results on the adversarial robustness. "Query Number" denotes the number of queries the attack system made to the target model and a higher number indicates more difficulty.

| Metric (↑ better) | ALBERT | | | + Shallow-Deep [5] | | | + PABEE *(ours)* | | |
|---|---|---|---|---|---|---|---|---|---|
| | SNLI | MNLI-m/-mm | Yelp | SNLI | MNLI-m/-mm | Yelp | SNLI | MNLI-m/-mm | Yelp |
| **Original Acc.** | 89.6 | 84.1 / 83.2 | 97.2 | 89.4 | 82.2 / 80.5 | 97.2 | **89.9** | **85.0 / 84.8** | **97.4** |
| **After-Attack Acc.** | 5.5 | 9.8 / 7.9 | 7.3 | 9.2 | 15.4 / 13.8 | 11.4 | **19.3** | **30.2 / 25.6** | **18.1** |
| **Query Number** | 58 | 80 / 86 | 841 | 64 | 82 / 86 | 870 | **75** | **88 / 93** | **897** |

drops as the inference speed goes up, PABEE has an inverted-U curve. We confirm this observation statistically with Monte Carlo simulation in Appendix C. To analyze, when the patience $t$ is set too large, the later internal classifier may suffer from the overthinking problem and make a wrong prediction that breaks the stable state among previous internal classifiers, which have not met the early-exit criterion because $t$ is large. This makes PABEE leave more samples to be classified by the final classifier $C_n$, which suffers from the aforementioned overthinking problem. Thus, the effect of the ensemble learning vanishes and undermines its performance. Similarly, when $t$ is relatively small, more samples may meet the early-exit criterion by accident before actually reaching the stable state where consecutive internal classifiers agree with each other.

**Impact of Model Depth** We also investigate the impact of model depth on the performance of PABEE. We apply PABEE to a 24-layer ALBERT-large model. As shown in Table 4, our approach consistently improves the accuracy as more layers and classifiers are added while producing an even larger speed-up ratio. This finding demonstrates the potential of PABEE for burgeoning deeper PLMs [44–46].

### 4.6 Defending Against Adversarial Attack

Deep Learning models have been found to be vulnerable to adversarial examples that are slightly altered with perturbations often indistinguishable to humans [47]. Jin et al. [6] revealed that PLMs can also be attacked with a high success rate. Recent studies [5, 27] attribute the vulnerability partially to the overthinking problem, arguing that it can be mitigated by early exit mechanisms.

In our experiments, we use a state-of-the-art adversarial attack method, TextFooler [6], which demonstrates effectiveness on attacking BERT. We conduct black-box attacks on three datasets: SNLI [48], MNLI [39] and Yelp [49]. Note that since we use the pre-tokenized data provided by Jin et al. [6], the results on MNLI differ slightly from the ones in Table 1. We attack the original ALBERT-base model, ALBERT-base with Shallow-Deep [5] and with Patience-based Early Exit.

As shown in Table 5, we report the original accuracy, after-attack accuracy and the number of queries needed by TextFooler to attack each model. Our approach successfully defends more than $3\times$ attacks compared to the original ALBERT on NLI tasks, and $2\times$ on the Yelp sentiment analysis task. Also, PABEE increases the number of queries needed to attack by a large margin, providing more protection to the model. Compared to Shallow-Deep [5], our model demonstrates significant robustness improvements. To analyze, although the early exit mechanism of Shallow-Deep can prevent the aforementioned overthinking problem, it still relies on a single classifier to make the final prediction, which makes it vulnerable to adversarial attacks. In comparison, since Patience-based Early Exit exploits multiple layers and classifiers, the attacker has to fool multiple classifiers (which may exploit different features) at the same time, making it much more difficult to attack the model. This effect is similar to the merits of ensemble learning against adversarial attack, discussed in previous studies [50–52].

# 5 Discussion

In this paper, we proposed PABEE, a novel efficient inference method that can yield better accuracy-speed trade-off than existing methods. We verify its effectiveness and efficiency on GLUE and provide theoretical analysis. Empirical results show that PABEE can simultaneously improve the efficiency, accuracy, and adversarial robustness upon a competitive ALBERT model. However, a limitation is that PABEE currently only works on models with a single branch (e.g., ResNet, Transformer). Some adaption is needed for multi-branch networks (e.g., NASNet [53]). For future work, we would like to explore our method on more tasks and settings. Also, since PABEE is orthogonal to prediction distribution based early exit approaches, it would be interesting to see if we can combine them with PABEE for better performance.

## Broader Impact

As an efficient inference technique, our proposed PABEE can facilitate more applications on mobile and edge computing, and also help reduce energy use and carbon emission [54]. Since our method serves as a plug-in for existing pretrained language models, it does not introduce significant new ethical concerns but more work is needed to determine its effect on biases (e.g., gender bias) that have already been encoded in a PLM.

## Acknowledgments and Disclosure of Funding

We are grateful for the comments from the anonymous reviewers. We would like to thank the authors of TextFooler [6], Di Jin and Zhijing Jin, for their help with the data for adversarial attack. Tao Ge is the corresponding author. The authors did not receive third-party funding or support for this work.

## Footnotes

*Equal contribution. Work done during these two authors' internship at Microsoft Research Asia.

[2]Code available at `https://github.com/JetRunner/PABEE`.

[3]https://www.quora.com/q/quoradata/First-Quora-Dataset-Release-Question-Pairs

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
