[Supplementary Material]

# A Image Classification

To verify the effectiveness of PABEE on Computer Vision, we follow the experimental settings in Shallow-Deep [5], we conduct experiments on two image classification datasets, CIFAR-10 and CIFAR-100 [55]. We use ResNet-56 [10] as the backbone and compare PABEE with BranchyNet [26] and Shallow-Deep [5]. After every two convolutional layers, an internal classifier is added. We set the batch size to 128 and use SGD optimizer with learning rate of $0.1$.

Table 6: Experimental results (median of 5 runs) of ResNet based models on CIFAR-10 and CIFAR-100 datasets.

| Method | CIFAR-10 | | CIFAR-100 | |
|---|---|---|---|---|
| | Speed-up | Acc. | Speed-up | Acc. |
| ResNet-56 [10] | $1.00\times$ | 91.8 | $1.00\times$ | 68.6 |
| BranchyNet [26] | $1.33\times$ | 91.4 | $1.29\times$ | 68.2 |
| Shallow-Deep [5] | $1.35\times$ | 91.6 | $1.32\times$ | 68.8 |
| PABEE *(ours)* | $1.26\times$ | **92.0** | $1.22\times$ | **69.1** |

The experimental results in CIFAR are reported in Table 6. PABEE outperform the original ResNet model by $0.2$ and $0.5$ in terms of accuracy while speed up the inference by $1.26\times$ and $1.22\times$ on CIFAR-10 and CIFAR-100, respectively. Also, PABEE demonstrates a better performance and a similar speed-up ratio compared to both baselines.

# B Proof of Theorem 1

**Proof B.1** *Recap we are in the case of binary classification. We denote the patience of PABEE as $t$, the total number of internal classifiers (IC) as $n$, the misclassification probability (i.e., error rate) of all internal classifiers as $q$, and the misclassification probability of the final classifier and the original classifier as $p$. We want to prove the PABEE mechanism improves the accuracy of conventional inference as long as $n - t < (\frac{1}{2q})^{t+1} p - q$.*

*For the examples that do not early-stopped, the misclassification probability with and without PABEE is the same. Therefore, we only need to consider the ratio between the probability that a sample is early-stopped and misclassified (denoted as $p_{misc}$) and that a sample is early-stopped (denoted as $p_{stop}$). We want to find the condition on $n$ and $t$ which makes $\frac{p_{misc}}{p_{stop}} < p$.*

*First, considering only the probability mass of the model consecutively output the same label from the first position, we have*

$$p_{stop} > q^{t+1} + (1-q)^{t+1} \tag{7}$$

*which is the lower bound of $p_{stop}$ that only considering the probability of a sample is early-stopped by consecutively predicted to be the same label from the first internal classifier. We then take its derivative and find it obtains its minimum when $q = 0.5$. This corresponds to the case where the classification is performing random guessing (i.e. classification probability for class 0 and 1 equal to 0.5). Intuitively, in the random guessing case the internal classification results are most instable so the probability that a sample is early-stopped is the smallest.*

*Therefore, we have $p_{stop} > (\frac{1}{2})^t$.*

*Then for $p_{misc}$, we have*

$$p_{misc} < q^{t+1} + (n-t-1)(1-q)q^{t+1} \tag{8}$$

*where $q^{t+1}$ is the probability that the example is consecutively misclassified for t+1 times from the first IC. The term $(n-t-1)(1-q)q^{t+1}$ is the summation of probability that the example is consecutively misclassified for t+1 times from the $2, ..., n-t$ th IC but correctly classified at the previous IC, without considering the cases that the the inference may already finished (whether correctly or not) before that IC. The summation of these two terms is an upper bound of $p_{misc}$.*

*So we need to have*

$$(n-t)q^{t+1} - (n-m-1)q^{t+2} < (\frac{1}{2})^t p \qquad (9)$$

*which equals to*

$$(n-t)(q^t - q^{t+1}) < (\frac{1}{2})^t (\frac{p}{q}) - q^{t+1} \qquad (10)$$

*which equals to*

$$n - t < \frac{(\frac{1}{2q})^t (\frac{p}{q}) - q}{1 - q} < (\frac{1}{2q})^t (\frac{p}{q}) - q \qquad (11)$$

$\blacksquare$

Specially, when $q = p$, the condition becomes $n - t < (\frac{1}{2p})^t - p$

## C  Monte Carlo Simulation

To verify the theoretical feasibility of Patience-based Early Exit, we conduct Monte Carlo simulation. We simplify the task to be a binary classification with a 12-layer model which has classifiers $C_1 \ldots C_{12}$ that all have the same probability to correctly predict.

(a) Accuracy lower bound of each single PABEE classifier to achieve the original accuracy. The translucent black plain denotes inference without PABEE.

(b) Accuracy requirement reduction effect of PABEE classifiers.

Figure 5: Monte Carlo simulation of per PABEE classifier's accuracy vs. the original inference accuracy under different patience settings.

Shown in Figure 5a, we illustrate the accuracy lower bound of each single $C_i$ needed for PABEE to reach the same accuracy as the original accuracy without PABEE. We run the simulation 10,000 times with random Bernoulli Distribution sampling for every 0.01 of the original accuracy between 0.5 and 1.0 with patience $t \in [1, 11]$. The result shows that Patience-based Early Exit can effectively reduce the needed accuracy for each classifier. Additionally, we illustrate the accuracy requirement reduction in Figure 5b. We can see a valley along the patience axis which matches our observation in Section 4.5. However, the best patience in favor of accuracy in our simulation is around 3 while in our experiments on real models and data suggest a patience setting of 6. To analyze, in the simulation we assume all classifiers have the same accuracy while in reality the accuracy is monotonically increasing with more layers involved in the calculation.