[Reviews · NeurIPS 2020]

Review 1

Summary and Contributions: The authors proposes early stopping at test-time to improve inference speed and accuracy. The idea is to train a classifier at each layer of multi-layered embedding model like BERT and perform classification one layer at time, stopping when the prediction stops changing. They demonstrate empirically that the method improves both the speed and accuracy of BERT/ALBERT on the GLUE benchmarks. My opinion of the work remains the same after the response.

Strengths: Simple straightforward idea that would be easy to implement directly from the description of the paper and that performs better in some cases than more complicated methods. The prevalence of these models means the idea could be pretty impactful, especially in industry where there is a heavy reliance on out-of-the-box methods like BERT and throughput and latency concerns really matter. The method could presumably be combined with other work like head pruning since it is somewhat orthogonal.

Weaknesses: The primary weakness is that the speed improvements are modest, only about 50\% faster. Another weakness is that it is specific to architectures with many layers, like 12-layer BERT. An architecture with MUSE, with less layers may have less opportunity for improvement.

Correctness: Yes. They use two popular embedding methods as base models (BERT/ALBERT) and provide results on the GLUE benchmark. While there are obviously problems with GLUE, they aren't simply trying to eke out every bit of SOTA performance, but instead demonstrating the viability of their method.

Clarity: Yes. Crystal clear.

Relation to Prior Work: This is a neat idea and I'm surprised it hasn't been done before. The relation to prior work is clear, but there are gaps in my own knowledge of prior work in this specific area. The method seems orthogonal to many existing techniques, like head pruning, which could allow it to be combined with them for even bigger improvements. As a remark, I'm not fully sure how similar this work is to citations, which were noted as doing something similar [28,29,30]. I didn't get a chance to read them carefully, but these works were very recently put on arxiv and so even if they are similar, in the spirit of independent discovery, I wouldn't hold it against the novelty of the work.

Reproducibility: Yes

Additional Feedback: I like the simplicity of this idea, and the connection to the 'overthinking' problem makes the idea deceptively simple and elegant. My main concern is that the speed improvements are somewhat modest, and in general are limited by the architecture depth and the extent of the over thinking problem. It would be interesting to see how well this works on a sentence embedding model like MUSE, which happens to have fewer layers. There might be an interesting connection to how humans perform classification. I'm reminded of a talk by Tom Mitchell in which he shows time-based scans of the human brain as it makes classification decisions. In it, some features would always be activated first (like is the object furry or not, possibly indicating a predator), indicating that the brain also does some type of early stopping for inference for decisions that must be made in a timely manner.


Review 2

Summary and Contributions: After author response: thanks for adding the suggested baselines and clarifications to the paper. === The paper introduces a simple approach for improving inference efficiency via early exists. Like past work, additional classification heads are added and trained at each layer. But unlike past work, the early exit criteria is based on the consistency of the classification decisions over several layers, rather than the entropy of the prediction distribution. The authors find that the proposed approach often improves end-task accuracy over the original baseline model (which makes predictions after all layers have been executed), an observation referred to by the authors as "overthinking." While I find some of the explanations for the observed behavior a bit unconvincing, the approach and empirical findings are very exciting. This work is the first I'm aware of that shows it's possible to simultaneously improve inference efficiency and end-task accuracy via early exits.

Strengths: This work is the first to show that it's possible to simultaneously improve inference efficiency and end-task accuracy via early exits. The experimental methodology is sound and support the paper's main *empirical* claims.

Weaknesses: While the empirical claims relating to the proposed approach are well supported, I'm not completely convinced by the theoretical explanations provided (e.g., the analogy between "overthinking" and overfitting, the theorem suggesting that the proposed approach is guaranteed to improve end-task accuracy). For instance, I find the analogy in Section 3.1 between overfitting and the observed "overthinking" phenomenon to be pretty unconvincing. The former is about generalization, while the latter is about the representations learned at different layers and their suitability to a particular input. Maybe this was the inspiration for your idea, but I don't think you've shown a strong enough connection to say that the two are analogous. I'm also unconvinced by Theorem 1, which seems to try to guarantee that the proposed early exit approach would improve end-task accuracy. This theorem seems to assume that the misclassification probability at different layers are independent of one another. This assumption should be stated more clearly, but also isn't a reasonable assumption in my opinion. Misclassifications at different layers are almost certainly not independent. Consider Pr(misclassified at layer i+1 | correctly classified at layer i) vs. Pr(misclassified at layer i+1 | misclassified at layer i). Are these equal? I suspect they are not, since some examples are inherently more challenging and will be misclassified at multiple successive layers. To my eyes, there are several alternative explanations for the observed behavior, for example: (1) different layers are comparatively better suited to some inputs than others, thus why early exits perform better; and/or (2) there's an ensembling effect from the proposed early exit criteria, which is why end-task accuracy improves compared to the baseline. These other explanations should ideally be considered and evaluated.

Correctness: The experiments are sound and support the paper's main *empirical* claims. However, the authors also provide a theoretical explanation for the reported results, and this explanation doesn't seem completely sound to me (see points above).

Clarity: The paper is very well written and easy to follow.

Relation to Prior Work: The comparison to related work is extensive, both quantitatively (comparing different approaches on the GLUE benchmark) and in words (describing how the proposed approach compares to other work on early exits).

Reproducibility: Yes

Additional Feedback: - The observation that PABEE can improve accuracy compared to the baseline (illustrated in Figures 3 and 4) is very interesting! I wonder if there's some kind of ensembling effect here. For example, with patience t=6, you're often picking the majority decision from k=12 intermediate classifiers. It'd be valuable to include another result in Table 1 or 2 showing the result from actually ensembling the outputs from all classification layers. This could potentially be an upper bound for end-task accuracy, but might help explain the observed accuracy improvement from PABEE. - In Figure 2b, could you please add error bars? The observed "overthinking" regime is subtle and could easily be due to random variation in finetuning. For example, Dodge et al. [1] found a standard deviation of 6.1% when finetuning PLMs on MRPC (Dodge et al., Table 3). The observed "overthinking" penalty appears to be smaller than that. - Please specify what evaluation metrics you report in Table 2. In particular, the GLUE benchmark sometimes uses F1 for MRPC/QQP and MCC for CoLA, but I'm guessing you use accuracy everywhere? - Why does training time decrease with PABEE in Table 3? Is it because early stopping happens earlier? References: [1] Dodge J, Ilharco G, Schwartz R, Farhadi A, Hajishirzi H, Smith N. Fine-tuning pretrained language models: Weight initializations, data orders, and early stopping.


Review 3

Summary and Contributions: In this paper the authors propose a new criterion for layer-wise early-stopping in multi-layer pre-trained language models. Like previous work, they add a classifier after each layer, and unlike previous work which uses confidence from the classifier to determine when to stop, their proposed approach is to stop when the same prediction is made by k layers, which they denote "patience-"based training. Their approach also directly applies to regression, unlike previous approaches designed for classification. This approach appears to out-perform alternative objectives in terms of accuracy, but not speed (not entirely clear), which is to be expected since by design their approach is likely to (must?) perform more computation before early-exit. Following author response: I'm sympathetic to the ACL papers being too recent, and in light of that and the rest of the response I've updated my review and score to "Marginally above the acceptance threshold."

Strengths: - A new criterion for early stopping in fine-tuned language models. The criterion appears to obtain better accuracy than others, though it remains unclear to me exactly how the speed-accuracy trade-off compares to previous work.

Weaknesses: - theoretical analysis is weak. they only consider (unrealistic) case where all internal classifiers have the same error rate, and the final classifier has the same error rate with and without internal classifiers. The former constraint is very unlikely to be true; see e.g. Figure 2b, where error rate decreases initially as a function of the number of layers. Personally I don't believe this analysis adds much insight to the paper, and certainly shouldn't be touted as a contribution. - could use more thorough comparison/discussion of previous work, and is missing some citations

Correctness: - it's not clear to me which baselines were implemented by the authors and which numbers are taken from existing papers. This makes it hard to tell how well they have compared to previous work. For example, as far as I can tell the numbers reported for DeeBERT in table 2 do not appear in the DeeBERT paper; they are lower than the numbers that appear in the paper. This makes me concerned that the baseline numbers reported in this paper are not fair/accurate. I'm also confused by your tables, which indicate that your model is more accurate but slower, and later plots of accuracy at a given speed-up, which claim that your model is faster at a given accuracy than previous work. If that is the case, then why aren't those numbers reflected in earlier tables comparing to previous work? This was clarified in author response, but should also be made more clear in the paper.

Clarity: The paper is reasonably well written, but could use a pass for clarity, including important experimental information, and checking equations. As explained above, important aspects of experimental results remain unclear.

Relation to Prior Work: Since there has been a recent deluge of papers on this exact topic (3 appearing at ACL), I would like to see much more detailed comparison to these approaches (related to my comments under the "correctness" setting). These papers are cited and compared against to some extent, but not thoroughly discussed. Also, it doesn't seem like the authors have read that work very carefully. In the related work section, they claim that Schwartz et al. 2020 employ the same approach as Kaya et al. 2019, when in fact, they do not. This further increases my concern that the empirical results aren't sound. This work is also related to previous work on cascades of increasingly complex models, e.g. the Viola-Jones cascade of classifiers for image recognition. Universal transformers (https://arxiv.org/abs/1807.03819) are also related, as is LayerDrop, which is cited in experimental results but not discussed in the related work section, and depth-adaptive transformers (https://arxiv.org/abs/1910.10073). I feel that the related work section is lacking, given that this is a fairly rich and long area of research. I think for example Figure 2 could be removed, making room for suggested improvements to related work and experimental results. After reading the author response, I am sympathetic to the fact that many of the highly related works are concurrent. I would still like to see a more detailed discussion of the similarities and differences, in the absence of time to perform empirical analysis.

Reproducibility: Yes

Additional Feedback: - personally I don't think it's beneficial to anthropomorphize neural network models, e.g. "patience", "decision-making process", "overthinking." I realize the term "overthinking" is taken from a highly related work, but you can still discuss the phenomenon without using the exact terminology, which vastly over-credits the network. - not sure that figure 2 is necessary - line 114: ensemble learning should not be capitalized - line 117: cross entropy should not be capitalized - In Figures 3 and 4, speed-up ratio is plotted in reverse order (increasing in one plot and decreasing in the other), which is confusing. - Please indicate what bold means in your tables. In some cases it appears to mean nothing, since the bolded number equals other numbers in the table. - you say "significantly out-performs" other models on e.g. line 197, but did you perform significance testing?


Review 4

Summary and Contributions: This paper proposes a method to improve efficiency during inference as well as reduce the problem of "overthinking" of deep models, by using classifiers that produce predictions from internal layers of a model, allowing early exit if the predictions of consecutive layers are consistent for greater than a predefined patience limit. Due to the combination of predictions from the internal classifiers, the paper claims that this approach has an ensembling effect which allows them to improve performance over having the full model.

Strengths: 1) Provides speed up during inference while also providing gains in performance over the full model baseline using ALBERT. 2) Compared to prediction score based approaches, has the advantage of being able to handle both classification and regression. 3) Dependence on the patience value t as well as model depths is explored. 4) Good results using state of the art ALBERT model. Good empirical evaluation, simple and straight froward approach which seems to work well.

Weaknesses: The performance gains are on the ALBERT framework where the internal layer weights are shared. Experiments on BERT do not show similar improvements which the authors conjecture is due to this discrepancy between the two models but needs to be examined further.

Correctness: Table 2, best results must be in bold which is from BERT base.

Clarity: Well written paper.

Relation to Prior Work: Missing reference: Universal transformer which has a dynamic halting mechanism. LayerDrop has been classified as a static approach but sub-networks of any desired depths can be extracted at inference time, so I believe it would better fit in the dynamic approach subsection.

Reproducibility: Yes

Additional Feedback:

[Author Response · NeurIPS 2020]

**General Response** Thanks for all reviewers for your insightful comments. We appreciate the reviewers for your
commendation for the simplicity, intuitiveness and effectiveness of our method. We will carefully address your
suggestions on typos, writing style and missing citations in the revision.

*About theoretical analysis:* The main contribution of our paper is a novel early exiting approach that empirically
performs well. The theoretical analysis has never been claimed to be the main contribution, instead it serves as a
supplementary analysis for trying to interpret the approach in a different aspect from the empirical analysis. Although it
is not with a tight constraint, as Reviewer 2 and 3 commented, we believe it is still meaningful to present the interesting
theoretical insights that help explain the observation of the empirical results (e.g., performance improvement), which
has been rarely discussed and studied by prior related work.

**Reviewer 1** Thanks for your endorsement and the pointer to an interesting neuroscience finding! As for the speed
improvement, our approach can improve the speed by around 50% while improving the accuracy on ALBERT-based
models. It can also improve the speed by around 150% or 250% with a moderate performance drop for a 12- or 24-layer
pretrained language models by decreasing the patience hyperparameter. It is true that our method works better with
deeper models. Considering recently released large pretrained language models like Megatron-LM and GPT-3 that
contain 72 and 96 layers respectively, we believe our approach will benefit future pretrained language models.

**Reviewer 2** *Response to Weaknesses:* (1) The similarity between overfitting and overthinking is indeed the motivation
behind our method. We acknowledge that we should better support this claim and will rewrite that part to make it more
convincing. (2) Please refer to "General Response". (3) Yes, we've also discussed the ensemble effect in our paper (Ln.
113-114) and "different layers are good for different inputs" is an interesting view point that we would like to explore
further.

*Response to Additional Feedback:* (1) It is indeed helpful and we will add this ensemble baseline in our revision. (2)
Yes, it seems smaller than overfitting. We will add error bars. (3) Following DistilBERT [Sanh et al., 2019], we report
the average of metrics (e.g., F1 and accuracy for MRPC). We will clarify this in the revision. (4) We are also surprised
by the training time reduction. Our guess is that training by taking multiple losses can facilitate the training process,
similar to multitask learning.

**Reviewer 3** *Response to Weaknesses:* Please refer to "General Response".

*Response to Correctness:* (1) As we described in the caption of Table 2, all results shown in Table 2 are medians of
5 runs. For DeeBERT, we use the official code to obtain the results **on the development set**. Note that the results
reported in the DeeBERT paper are on the test set, and are not median/average results. We have contacted the authors of
DeeBERT and got their official numbers on development set (81.65 on MNLI, 91.06 on SST-2). We will update our
paper with their results. The results of LayerDrop and HeadPrune are also reproduced with their officially released code.
For BranchyNet and ShallowDeep, we implement them by closely following the original papers. We will open-source
these two implementations along with PABEE. (2) Indeed, there is no contradiction between Table 1 and Figure 3.
We control all baselines to have a target speed-up between the speed of 6-layer and 9-layer ALBERT and report the
**highest accuracy** in Table 1. As shown in Figure 3, PABEE has a slightly lower acceleration ratio when achieving
its performance peak and this is the reason why PABEE in Table 1 looks slower than BranchyNet and Shllow-Deep.
Actually, Figure 3 shows that PABEE achieves a better speed-accuracy trade-off (better accuracy at a given speed-up
ratio) compared to the baselines across a wide acceleration spectrum. (3) **Equation 6 is correct**. As described in Ln.
121, we allocate more weights to later classifiers following [Kaya et al., 2019]. As stated in Section 3.2 of [Kaya et al.,
2019], this design choice is because: *"the earlier ICs have less learning capacity."*

*Response to Related Work:* (1) We would like to kindly point out that the proceedings of ACL are released **after the
NeurIPS deadline**. Also, these papers (including their arXiv preprints) were released within two months before the
NeurIPS deadline and thus considered concurrent work according to NeurIPS's policy. We have tried our best to cite
their arXiv preprints and add some discussion about them but it is not possible to have a thorough comparison, especially
given that two of them are **not evaluated on GLUE**. (2) [Schwartz et al., 2020] does use the same exiting criteria as in
[Kaya et al., 2019] (max prediction score) but they differ in details. We will rephrase this sentence. (3) Thanks for the
pointers to more related work! We will add them in the revision.

**Reviewer 4** Thanks for your endorsement and we are glad that you like the simplicity of our method!

*Response to Weaknesses:* We will run an experiment on pretraining BERT with the random layer numbers (in the same
way we fine-tune on downstream tasks) to verify our guess that the mismatch between pretraining and fine-tuning
causes that.

*Response to Related Work:* Thanks for pointing out the missing citation. We categorized LayerDrop as a static approach
because it specifies a predefined set of layers to be used during inference, so the number of layers to be used is not
dynamically adjusted with respect to the input. We will reconsider the categorization of LayerDrop per your suggestion.

[Meta-Review · NeurIPS 2020]

The authors proposes early stopping at test-time to improve inference speed and accuracy. The idea is to train a classifier at each layer of multi-layered embedding model like BERT and perform classification one layer at time, stopping when the prediction stops changing. The paper proposes a simple but effective technique that leads to good empirical results. Most excitingly, the authors report both speed-ups & higher accuracy. There are some concerns about the presented theorem, since the main assumption is unrealistic: it is assumed that the misclassification probabilities at different layers are independent of one another. This assumption should be stated more clearly, but also isn't very reasonable since misclassifications at different layers are almost certainly not independent. All in all, the theorem is not a central part of the paper, which is primarily an empirical paper. It could be worth rethinking some of the presentation as several of the reviewers were not convinced by the discussion of "overthinking". Finally, while it was not possible/needed to discuss concurrent ACL work in the submission, the AC suggests that it is discussed in the final version since ACL papers have now been out for a while.